# Characteristics of Work-Related Musculoskeletal Disorders in Korea

**DOI:** 10.3390/ijerph20021024

**Published:** 2023-01-06

**Authors:** Dohyung Kee

**Affiliations:** Department of Industrial Engineering, Keimyung University, Daegu 42601, Republic of Korea; dhkee@kmu.ac.kr; Tel.: +82-53-580-5319

**Keywords:** work-related musculoskeletal disorder, low back pain, rotator cuff injury, industrial accident, occupational disease

## Abstract

This study aimed to analyze trends for work-related musculoskeletal disorders (WMSDs) from 1996 to 2020 in Korea and to investigate characteristics of WMSDs, including WMSD approval rates, distribution by WMSD names, and the effects of industry type and size, and workers’ age and gender on WMSD occurrence. The data included those obtained from the official yearbooks for industrial accidents published by the Ministry of Employment and Labor and those obtained personally from the Korea Workers’ Compensation & Welfare Service. The results showed that although the incidence of WMSDs differed by year, approximately 9500 cases of WMSDs occurred in 2019 and 2020, the incidence rate of WMSDs was approximately 5.0 per 10,000 workers, and the proportions of WMSDs among industrial accidents were almost 9%. Low back pain was the leading cause of WMSDs; WMSDs occupied 9.5–71.5% of total occupational diseases by year and occurred most frequently in the manufacturing industry, followed by construction, transportation/warehouse and communication, and mining industries, and nearly 60% of WMSDs occurred in small business with <50 workers. Among chronic WMSDs, rotator cuff syndrome in the shoulder ranked first, intervertebral disc disorders second, and rotator cuff and tendon injuries third. By body parts, the shoulder was most susceptible to chronic WMSDs, followed by the low back, leg, and elbow/lower arm. The chi-square test and logistic regression analysis showed that industry type and size and workers’ gender and age were significantly associated with WMSD approval. It can be concluded that the WMSD preventive efforts should focus on low back pain and rotator cuff syndrome by WMSD name, manufacturing by industry, small business by industry size, men by gender, and aged workers by age.

## 1. Introduction

Musculoskeletal disorders (MSDs) are the leading contributor to disability worldwide, with low back pain being the single leading cause of disability in 160 countries. The disability associated with MSDs has been increasing and is projected to rapidly increase in the next decades because of population increase and aging [1]. MSDs are considered to be work-related musculoskeletal disorders (WMSDs) if an event or exposure in the work environment either caused or contributed to the resulting disorders or significantly aggravated pre-existing disorders. In the USA, 272,780 WMSD cases were reported in 2018, with the incidence rate of WMSD cases being 27.2 per 10,000 full-time workers, accounting for approximately 30% of all occupational injuries and illnesses involving days away from work [2]. In Korea, 9601 cases of WMSDs were reported in 2020, of which low back pain comprised approximately 43.5% (4177 cases) [3].

WMSDs constitute one-third of work injuries and illnesses. They have significant economic and social consequences, including rising costs of wage compensation and medical expenses, reduced productivity, and lower quality of life [4,5,6]. WMSDs cost 40% of global compensations for occupational and work-related injuries and diseases, and in 2005, they took up 59% of all recognized occupational diseases across the European Union [7].

For establishing efficient preventive measures or policies for WMSDs, studies dealing with WMSD characteristics based on real WMSD case data, such as effects of workers’ age, industry type and size, MSD name, etc., are needed. However, most previous studies focused on the prevalence of WMSDs based on subjective questionnaire surveys, the liabilities of which may not be high [8,9,10,11,12,13]. For example, Iglesias et al. [14] reported a significant prevalence of musculoskeletal complaints in daily podiatry work. Rahimi et al. [15] pointed out that the prevalence of WMSDs was 94% in Iranian physiotherapists. Both studies were based on the Standardized Nordic Questionnaire. Some researchers studied the prevalence of WMSDs among the workers in motor companies, ship building, subway train repair, disabled infant and children care center workers, dentists, nurses, and agricultural workers in Korea, all of which were based on subjective questionnaires [16,17,18,19,20,21,22,23]. The Rural Development Agency (RDA), an affiliation of the Ministry of Agriculture, Food and Rural Affairs of Korea, investigated diseases of agricultural, forestry, and fishing workers aged 19 or more in 10,020 households in 2020, which was based on a direct interview survey using a structured questionnaire. WMSD was identified as the most prevalent disease (84.6%), followed by circulatory system diseases (3.0%), skin diseases (2.9%), and nervous system diseases (2.1%) [23].

The Ministry of Employment and Labor of Korea publishes an annual data book for industrial accidents categorized by total industrial accidents, including accidents and occupational diseases, fatal industrial accidents, and occupational diseases. The data book presents statistics for WMSD cases by disease name, industry type and size, and workers’ gender and age. It classifies WMSDs into three representative diseases: WMSDs caused by (1) bodily burdensome work, (2) chronic and accidental low back pain, and (3) carpal tunnel syndrome. Here, WMSDs caused by bodily burdensome work refer to all musculoskeletal disorders except for low back pain and carpal tunnel syndrome. Although the book contains statistics for real WMSD cases by industry type and size, and workers’ gender and age, it does not provide a more detailed status of WMSDs, such as specific WMSD names and susceptible body parts. In other words, there is no information on which WMSDs occurred frequently or what parts of the body were vulnerable to WMSDs.

Therefore, this study aimed to analyze the trends for real WMSD data from 1996 to 2020 using annual data books for industrial accidents by the Ministry of Employment and Labor of Korea, which include general statistics for WMSD cases, and to investigate the characteristics of WMSDs, including WMSD approval rates, distribution by WMSD names, effects of industry type and size, and workers’ age and gender on WMSD occurrence, based on personally obtained data from the Korea Workers’ Compensation & Welfare Service (COMWEL).

## 2. Materials and Methods

### 2.1. Data Collection

Data of WMSD cases were obtained from official yearbooks for industrial accidents published by the Ministry of Employment and Labor [3]. These yearbooks were used as data source for WMSDs because only the books provide official statistics for industrial accidents including WMSDs in Korea. The data include the number of industrial accidents and occupational diseases, including WMSDs, cardio- and cerebrovascular diseases, etc., classified by various factors, such as industry type and size, workers’ gender and age, working period, managerial and direct causes, injury type, injured body part, etc. For investigating various trends of WMSDs, the incidence rates of WMSDs per 10,000 workers, proportions of WMSDs among industrial accidents, rates of low back pain among WMSDs, and distribution of traditional occupational diseases, cardio- and cerebrovascular diseases, and WMSDs among total occupational diseases were calculated based on the data of the yearbooks in this study. Here, WMSDs are classified by industry type and size, and workers’ gender and age. For surveying the overall trends of WMSDs by year, the data on WMSDs from 1996 to 2020 were analyzed in this study, because the data have been collected and published annually by the Ministry of Employment and Labor since 1996, and the data for 2021 have not been published yet.

The characteristics of WMSDs in Korea, including WMSD approval rates, distribution by WMSD names, and effects of industry type and size, and workers’ age and gender on WMSD occurrence were investigated based on the data personally acquired from COMWEL through the information disclosure system. According to the information disclosure system, when a national or a foreigner requests information related to public institutions, the relevant public institutions shall disclose to the public the information that the institutions produce, hold, and manage while conducting business. The system is based on the Act on Information Disclosure of Public Institutions of Korea. The data were allowed to be used for research purposes only. COMWEL is a quasi-governmental agency under the Ministry of Employment and Labor, which implements various social security and labor welfare services and programs. In Korea, workers with a doctor’s certificate for MSDs can apply to COMWEL for medical treatment for these disorders. COMWEL determines whether the applicants’ MSDs are work-related by deliberation. If COMWEL judges the applicants’ MSDs to be work-related (i.e., WMSDs), it pays the person’s wage for the leave of absence and the cost of treatment during the medical care of the applicants.

Data for WMSD approval rates for the last six years (2016–2021) and individual workers’ applications for medical care attributed to industrial accidents for the last three years (2019–2021) were obtained from COMWEL. A total of 31,173 applications (9524, 10,000, and 11,649 applications for 2019, 2020, and 2021, respectively) used for deliberation were acquired and used in the following analyses. The data for 2019 and 2020 were part of the data for WMSDs in the yearbooks announced by the Ministry of Employment and Labor, and the data for 2021 have not yet been published officially. These were different from the data from the Ministry of Employment and Labor because COMWEL did not deliberate on accidental MSDs, such as accidental low back pain, rotator cuff tear, sprain, etc. The accidental MSDs were approved as WMSDs without deliberation by COMWEL only if the accidents that caused the MSDs were proven. In other words, the data acquired from COMWEL contained only chronic or degenerative MSDs. An individual application taken from COMWEL contained information on the applicant’s gender and age, industry type and size, WMSD name, and whether the application was approved as WMSDs.

### 2.2. Data Analysis

The chi-square test and logistic regression analysis were adopted to investigate the significance of the effects of applicants’ variables, such as industry type and size, workers’ age and gender on WMSD occurrence, and the relationships between the applicants’ variables and the WMSDs, respectively. All statistical analyses were conducted using SAS (SAS Inc., Cary, NC, USA) and Microsoft Excel (Microsoft Co., Redmond, WA, USA).

## 3. Results

### 3.1. WMSDs by Year

The Ministry of Employment and Labor began to collect and announce WMSD cases in Korea in 1996. The trend for WMSDs by year was similar to that of total occupational diseases (Figure 1). WMSD cases were less than 510 from 1996 to 1999 and have exceeded 1000 since the year 2000. The cases rapidly increased from 1999 to 2003 and consecutively decreased for two years after 2003. After 2007, the cases continuously decreased for four years and showed a flat level from 2011 to 2017. After 2017, there was a rapidly increasing trend again.

The incidence rates of WMSDs per 10,000 workers by year are presented in Figure 2. The rates varied from 0.16 in 1998 to 7.54 in 2004. For the last two years, the rates were approximately 5.0 per 10,000 workers, an increase of approximately 43% compared to the previous year.

The proportions of WMSDs among industrial accidents by year exhibited almost the same trend as the incidence rates of WMSDs per 10,000 workers (Figure 3). The proportions ranged from 0.2% to 8.9%, with an average of 4.9% (standard deviation: 2.8%). The proportions were almost 9% in 2019 and 2020.

The rates of low back pain among WMSDs ranged from 28.2% to 82.0% (Figure 4). The rate soared from 28.2% in 2004 to 82.0% in 2007 and was higher than 70.0% from 2006 to 2012. The ratios have steadily declined from 2012 and have been approximately 45% in the last three years.

The distribution of traditional occupational diseases, cardio- and cerebrovascular diseases, and WMSDs among total occupational diseases are illustrated in Figure 5. Here, the traditional occupational diseases include hearing loss, pneumoconiosis, organic solvent poisoning, etc., which frequently occur in the early stages of industrialization where the safety and health management was neglected, and decline as industrialization progresses. Occupational diseases collectively refer to three diseases of traditional occupational diseases, cardio- and cerebrovascular diseases, and WMSDs. WMSDs occupied 9.5–71.5% of occupational diseases by year. The ratio peaked at 71.5% in 2009 and decreased until 2017, accounting for 56.6% in 2017. For the last three years (2018–2020), the ratios of WMSDs have been approximately 65%.

### 3.2. WMSDs by Industry Type and Size and Workers’ Gender and Age

The distribution of WMSDs by industry for the last five years (2016–2020) is presented in Figure 6a. The Ministry of Employment and Labor classifies the industries into ten categories in yearbooks for industrial accidents, including mining, manufacturing, electricity/gas/vapor/water service, construction, transportation/warehouse and communication, forestry, fishery, agriculture, finance/insurance, and others. The WMSD cases in the electricity/gas/vapor/water service, forestry, fishery, agriculture, and finance/insurance were merged into ‘others’ because the annual cases in each industry were <40. WMSDs occurred most frequently in the manufacturing industry (42%) during 2016–2020, followed by construction (14%), transportation/warehouse and communication (5%), and mining (3%).

Small businesses with the number of workers of <100 occupied two-thirds of WMSDs reported for the last five years (68%) (Figure 6b). More than half of WMSDs occurred in small businesses with <50 workers (60%), while 13% of WMSDs occurred in large companies with 1000 or more workers.

The percentage or number of WMSD cases was almost 3.5 times higher in men (78%) than that in women (22%) (Figure 6c). Workers aged > 50 years accounted for more than half of total WMSDs (57%), while young and middle-aged workers < 40 years occupied less than 20% of WMSDs (19%) (Figure 6d).

### 3.3. WMSD Approval Rates

As stated above, COMWEL determines whether or not MSDs of applicants were attributed to work-related factors (i.e., MSDs approved (WMSDs) vs. MSDs rejected (MSDs not approved as WMSDs)). The number of applications for the six years (2016–2021) ranged from 8715 to 16,441 with a steady increase except in 2017 (Figure 7). The approval rate that the applicants’ MSDs were accepted as WMSDs rose by approximately 18%, from 54.0% in 2016 to 71.9% in 2019, and since then, it has slightly decreased to 66.6% in 2021.

### 3.4. WMSDs by Body Parts and Diseases

The distribution of WMSDs (approved MSDs by COMWEL) for the last three years of 2019–2021 by body parts is depicted in Figure 8. WMSDs occurred most frequently in the shoulder (38%), followed by the low back (25%), leg (12%), elbow/lower arm (10%), wrist/hand (7%), and neck (6%). WMSDs frequently occurred by body parts (≥50 cases) and their number of occurrences for the last three years are summarized in Table 1. By body parts, cervical intervertebral disc disorders (840 cases) occurred most frequently in the neck, rotator cuff syndrome (4927 cases) in the shoulder, medial epicondylitis (1277 cases) in the elbow/forearm, carpal tunnel syndrome (561 cases) in the wrist/hand, intervertebral disc disorders except accidental cases (3397 cases) in the lower back, and meniscus disorders/tears (1258 cases) in the leg. There was only a single WMSD with 50 or more cases in the upper arm (muscle and tendon rupture/injury: 305 cases). The number of cases for a WMSD regardless of body parts was ranked as follows: (1) rotator cuff syndrome in the shoulder (4927 cases); (2) intervertebral disc disorders in the low back (3377 cases); (3) rotator cuff and tendon injuries in the shoulder (1339 cases); (4) medial epicondylitis in the elbow/lower arm (1277 cases); (5) meniscus disorders/tears in the leg and foot (1258 cases); and (6) lateral epicondylitis in the elbow/lower arm (1182 cases).

### 3.5. Effects of Industry Type and Size and Workers’ Gender and Age on WMSDs

The chi-square test revealed that the industry type and size, and workers’ gender and age were significant on the occurrence of WMSDs (*p* < 0.01), based on the 31,173 applications obtained from COMWEL. The logistic regression analysis was adopted to quantify the relationship between WMSDs and industry type and size, and workers’ gender and age. In the analysis, the industry type and size, and workers’ gender and age, which were significant in the chi-square analysis, were used as independent variables, and whether the applicants’ MSDs were approved to be work-related (i.e., WMSDs vs. rejected MSDs) was employed as a dependent variable. Four logistic regression analyses were conducted for each independent variable (Table 2).

The industry was classified into five categories considering the number of WMSDs: mining, manufacturing, construction, transportation/warehouse and communication, and others. To increase the reliability of the analyses, industries with a small number of WMSDs (<50 WMSD cases per year) were merged into ‘others’. The odds ratios for industry type were calculated, with reference to industry of transportation/warehouse and communication, which had the lowest odds ratio. Construction had the highest odds ratio, followed by mining and manufacturing. Construction had an almost 3.5 times the probability that an applied MSD would be approved as work-related compared to the probability for the transportation/warehouse and communication industry.

The industry size was grouped into five to enhance the reliability of the logistic regression analysis, based on the Ministry of Employment and Labor yearbook: <49, 50–99, 100–299, 300–500, and ≥500. The logistic regression analysis for the industry size was performed with reference to sizes 50–99. Only large companies with ≥500 workers showed a significantly higher odds ratio than companies with 50–99 workers.

The relationships between WMSDs and workers’ characteristics, such as gender and age, were also analyzed. Men almost doubled the probability that an MSD would be approved as work-related (1.59) compared to the probability for women. The odds ratio gradually increased from the age of 20s and reached the highest value of 3.33 in the age of 50s. The odds ratios for age > 30s except for 70s (2.14–3.10) were almost 2–3 times higher than that of 20s. The odds ratio decreased after the age of 50s. The percentage concordant values of logistic regression models for the industry type and size, and workers’ age were slightly higher (42.2%, 36.5%, and 44.2%, respectively), while the value for the workers’ gender was relatively low (23.4%).

To investigate the effects of applicants’ industry size, gender, and age in each industry, logistic regression analyses by industry were conducted. The analyses revealed that the odds ratios of industry size with reference to 50–99s were larger in mining (<49: 2.68, 100–299: 0.93, 300–500: 2.04, and ≥500: 2.35) than in the overall industry in Table 2. The gender ratios with reference to women were larger in mining (1.89) and construction (2.31) than in the overall industry (1.59), and those of age with reference to 20s were larger in manufacturing (30s: 2.75, 40s: 3.80, 50s: 4.07, 60s: 3.80, and >70: 1.64) and construction (30s: 1.56, 40s: 3.32, 50s: 4.93, 60s: 6.68, and >70: 6.80).

## 4. Discussion

This study investigated the trends of WMSDs by year, industry type and size, and workers’ gender and age in Korea, based on the yearbooks for industrial accidents in Korea. In addition, the incidence rates of WMSDs per 10,000 workers, the proportion of low back pain among WMSDs, and the ratios of WMSDs among industrial accidents and total occupational diseases by year were calculated based on the data provided by the yearbooks. They were examined from 1996, when WMSDs were compiled as official statistics in Korea, to 2020 to grasp the overall trend of WMSDs. WMSDs by industry type and size, and workers’ gender and age were summarized as average values over the past five years (2016–2020) because recent trends are more significant. The distribution of WMSDs by body parts, specific diseases, and the effects of industry type and size, and workers’ gender and age on WMSDs were also analyzed based on the deliberation results of applications for WMSD medical care obtained from COMWEL. To the best of author’s knowledge, no study has dealt with the effects of industry type and size and workers’ gender and age on WMSDs, based on real MSD data rather than subjective symptoms. WMSD cases steadily increased from 1999 to 2003; this increment jumped rapidly in 2002 and 2003. In response, the Korean government enacted a law prescribing employers’ duty to prevent WMSDs in 2002, effective in July 2003 [24]. WMSDs dropped sharply from 2004, a year after the legal system related to WMSDs became effective, for two consecutive years. WMSDs increased again from 2006 to 2007 when accidental low back pain became newly included in WMSDs.

The incidence of WMSDs remained stagnant at about 5000–5500 cases in the early to mid-2010s, but has rapidly increased since 2018. This may be mainly due to the relaxation of the government’s WMSD recognition standards, not the deterioration of the working environment. The Ministry of Employment and Labor of Korea introduced the presumptive principle for six frequently occurring WMSDs, including cervical and lumbar disc herniations, rotator cuff syndrome, carpal tunnel syndrome, epicondylitis, and meniscus disorder/tear, to shorten the processing period for industrial accidents. Accordingly, if the requirements prescribed in the presumptive principle are met, the applicants’ MSDs are recognized as WMSDs without deliberation by COMWEL. For example, an applicant’s rotator cuff syndrome is approved as a WMSD as long as the following requirements are met: (1) an orthopedic surgeon confirmed the morbidity and illness, (2) the applicant was engaged in one of the following jobs: construction, transportation, cleaning, automobile manufacturing, shipbuilding, food industry, or tire and rubber product manufacturing, and (3) the applicant has been worked for ≥9 years [25]. The increase in WMSD cases is proved by the fact that the WMSD approval rate sharply increased by 8.5% in 2018 (70.0%) compared to 2017 (61.5%) (Figure 7).

The incidence rate of WMSD cases per 10,000 workers in Korea was much lower than that in the USA. While the rate in the USA was 27.2 per 10,000 full-time workers in 2018 [2], the rate in Korea was 5.1 per 10,000 workers in 2020. However, the working environment in Korea may not be safer than that in the USA. This may mean that the occurrence of WMSDs might likely increase in Korea, compared to the high incidence rate of WMSDs in the USA. Therefore, preventive efforts for reducing WMSDs by the Korean government and industries are needed.

Although the proportion of low back pain among occupational diseases has steadily decreased in recent years, low back pain was the most frequently occurring WMSD, accounting for 43–55% of WMSDs in Korea. This agrees with the previous study reporting that low back pain is the single leading WMSD [1,14,15]. Although rotator cuff syndrome was the most common WMSD recognized by COMWEL (Table 1), low back pain, including chronic and accidental cases, ranked first in terms of the number of occurrences (there was only chronic low back pain under deliberation by COMWEL in Table 1). This may imply that WMSD preventive measures should focus on low back pain and rotator cuff syndrome.

WMSDs occupied approximately two-thirds (56–65%) of all occupational diseases in Korea for the last five years (2016–2020), while traditional occupational diseases were approximately 25–35%. This large proportion of WMSDs may be because the number of traditional occupational diseases has steadily decreased due to the progress of industrialization and the strengthening of the safety and health management system with relevant regulations and laws. For example, in 2020, the number of workers in the mining industry, one of the most hazardous industries where traditional diseases such as hearing loss, pneumoconiosis, etc., are more frequent, decreased by about 53% compared to 1996, while the number in the manufacturing industry, which experiences frequent WMSDs due to stressful working postures, high motion repetition, static posture, vibration, extreme temperature, etc. [26,27,28,29], increased by about 41% [30]. This may also suggest that efforts to prevent occupational diseases should focus on preventing WMSDs, especially in the manufacturing industry.

More than three-fourths of WMSDs occurred in men over the last five years (2016–2020). This may be partly attributed to the fact that (1) the number of men who applied to COMWEL for medical care was more than that of women (in 2021, the number of men applicants was 8919 (76.6%), while that of women was 2730 (23.4%)), and (2) the proportion of employed men was higher (61.3%) than that of women (38.7%) [30]. Therefore, it can be judged that the incidence rate of WMSDs is higher in men than in women. This agrees with the result of the logistic regression analysis that the probability that an MSD would be approved as work-related was higher in men (1.59) than in women. However, this is not in agreement with Iglesias et al.’s finding [14], which reported more musculoskeletal complaints in female podiatrists.

Over half of WMSDs (57%) occurred in workers aged > 50 in Korea. This disagrees with the number of WMSD cases by age group in the USA in 2018, which showed that WMSDs occurred almost evenly across all age groups from 25 to 64 years [2]. This also does not agree with Iglesias et al.’s study [14], in which younger podiatrists showed more musculoskeletal complaints.

The chi-square tests and logistic regression analyses showed that workers’ age was significantly associated with WMSD approval. The high WMSD approval rate may be interpreted as the high prevalence of WMSDs. In addition, although applicants’ age is not the same as their length of service, it can be assumed that age is proportional to the length of service. Therefore, it can be inferred from this that the length of service is associated with WMSDs. The above findings disagree with the findings of Collins and O’Sullivan [31], Kee and Seo [18], Lusted et al. [32], Smith et al. [10,11], Widanarko et al. [33], and Yip [13]. These studies reported that age and working experience were not significantly related to the prevalence rates of WMSDs in nursing. In contrast, the present study agrees with the results of Dianat et al. [34], Heiden et al. [35], and Vieira et al. [36], which pointed out that age, gender, and length of service were associated with WMSD symptom prevalence.

In the logistic regression analyses, the larger odds ratios for industry size in mining may be attributed to the characteristics of the hazardous mining industry compared to other industries. In other words, the larger odds ratio indicates that mining is more hazardous than other industries. The larger odds ratio for men working in mining and construction industries may be because more men were employed in mining and construction than women (in 2019, men: 88.9% and 86.0% of workers in mining and construction, respectively) [30], and men might be generally exposed to more stressful tasks than women.

Although this study described various trends and characteristics of WMSDs in Korea, based on real WMSD or MSD data, it has some limitations. The data obtained from COMWEL did not contain data on stature, body weight, length of service, daily work hours, tasks performed, and disease history of the individuals who applied for WMSD medical care. Some studies showed that these variables significantly affected the prevalence of WMSDs or vice versa [10,11,17,18,31,32,33,34,35,36]. In addition, COMWEL provided MSD data for just three years (2019–2021). As a result, it was impossible to investigate the effects of various variables related to individual, task, and organization on WMSDs for longer periods. Therefore, further studies are required to overcome this study’s limitations using more data with additional variables for the applicants and organizations.

## 5. Conclusions

This study presented various trends of WMSDs by year, distribution of WMSDs by body parts and their names, and the effects of industry type and size and workers’ gender and age on WMSDs. The study findings could be used as basic data to understand the trends and characteristics of WMSDs and to establish preventive measures for WMSDs in Korea. For example, the high proportion of WMSDs in total occupational diseases might imply that preventive measures for reducing occupational diseases should focus on the prevention of WMSDs. In turn, the WMSD preventive efforts by the government and industries should focus on low back pain and rotator cuff syndrome by WMSD name, manufacturing by industry, men by gender, and aged workers by age.

## Figures and Tables

**Figure 1 ijerph-20-01024-f001:**
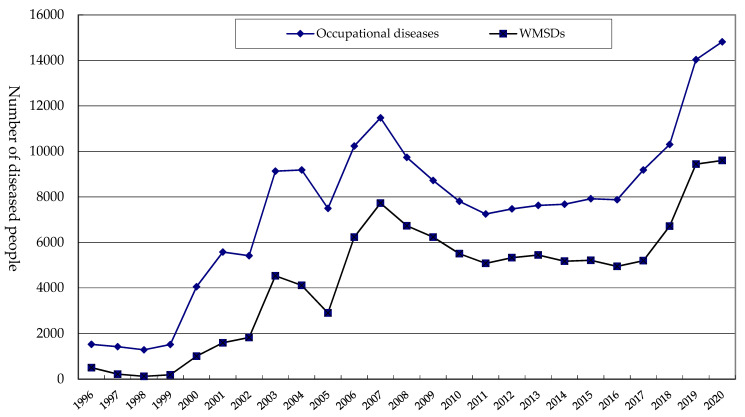
Trends for occupational diseases and WMSDs by year.

**Figure 2 ijerph-20-01024-f002:**
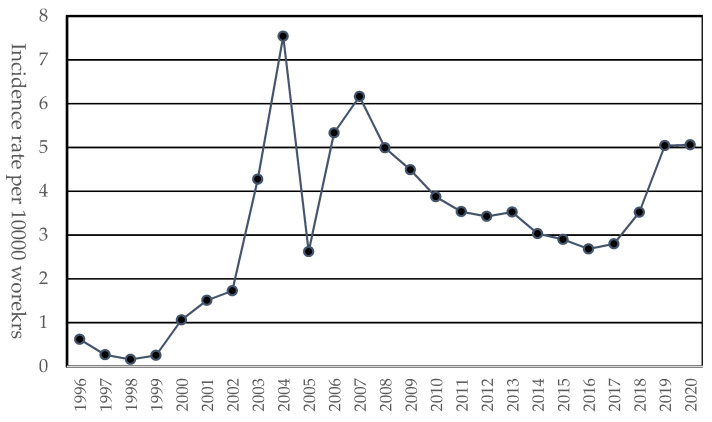
Incidence rates of WMSDs per 10,000 workers by year.

**Figure 3 ijerph-20-01024-f003:**
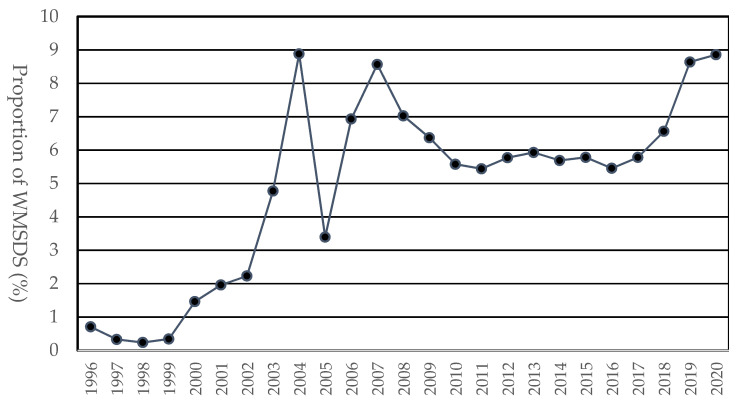
Proportions of WMSDs among industrial accidents by year.

**Figure 4 ijerph-20-01024-f004:**
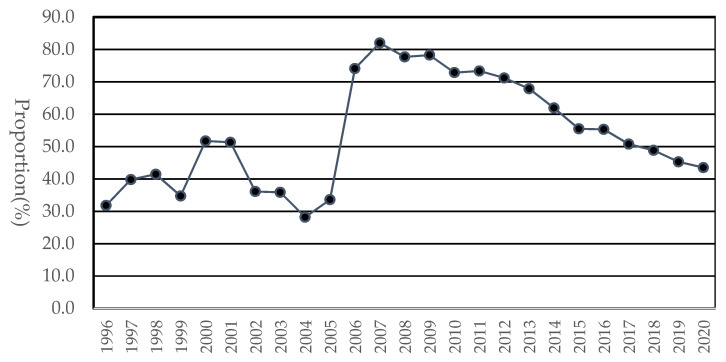
The proportion of low back pain among WMSDs by year.

**Figure 5 ijerph-20-01024-f005:**
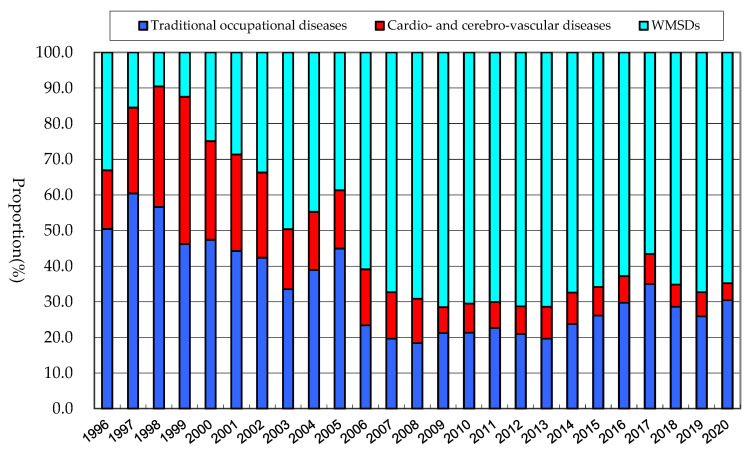
Distribution of WMSDs among occupational diseases by year.

**Figure 6 ijerph-20-01024-f006:**
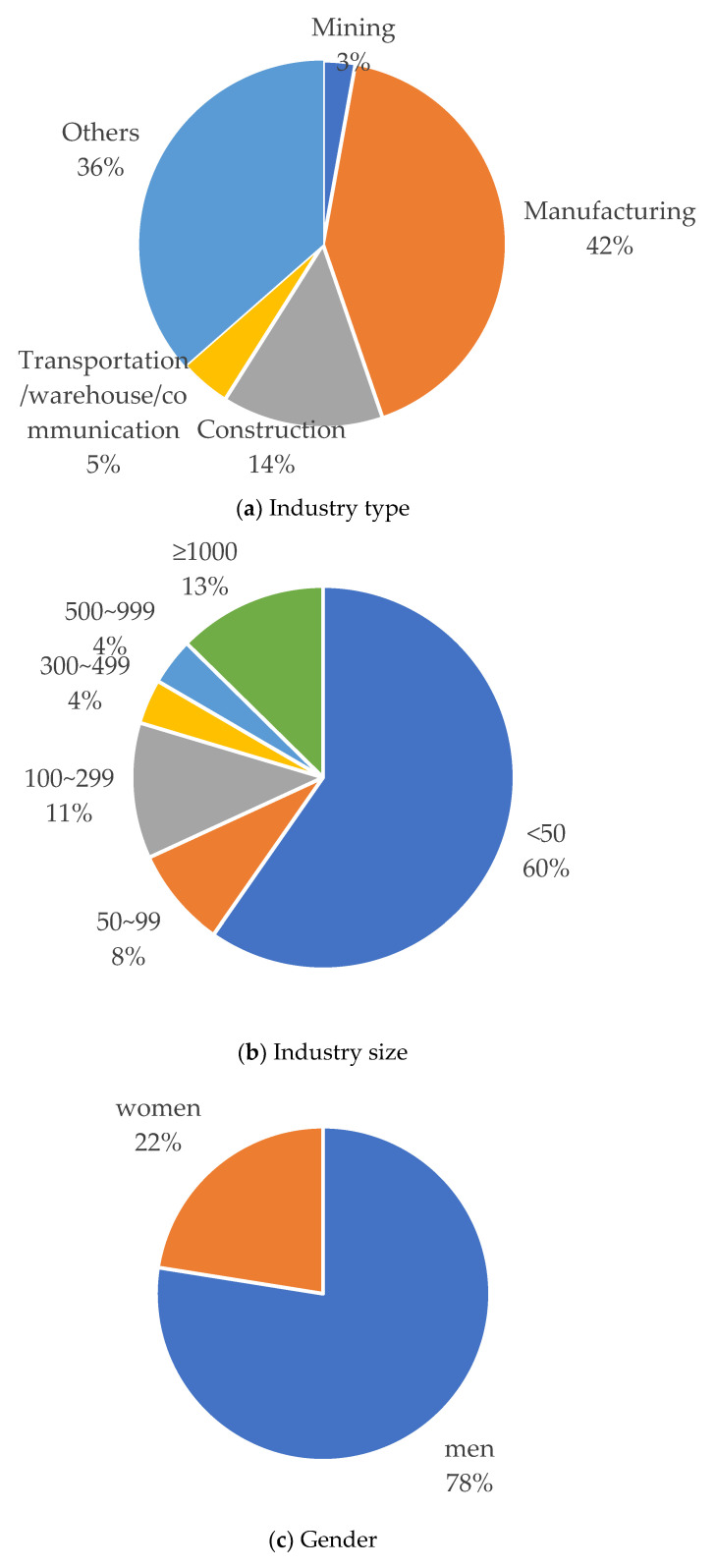
Distribution of WMSDs by industry type and size and workers’ gender and age.

**Figure 7 ijerph-20-01024-f007:**
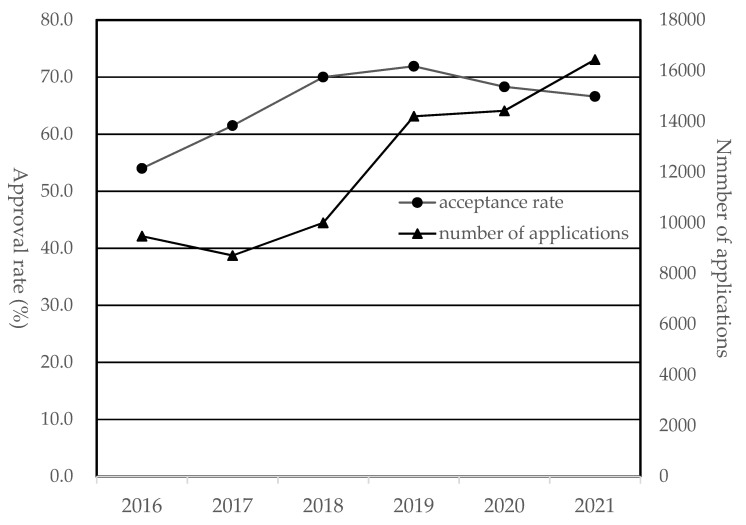
Number of applications and WMSD approval rates by year.

**Figure 8 ijerph-20-01024-f008:**
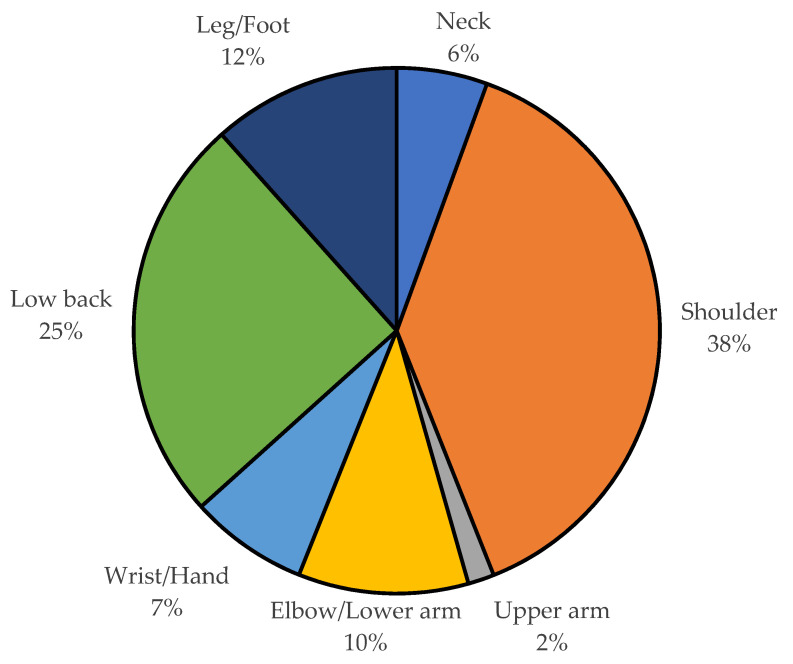
Distribution of WMSDs by body parts.

**Table 1 ijerph-20-01024-t001:** WMSDs by body parts and names.

Body Part	WMSD Name	Number of Cases
Neck	Cervical intervertebral disc disorders	840
	Herniation of cervical intervertebral disc	127
	Spinal stenosis	77
Shoulder	Rotator cuff syndrome	4927
	Rotator cuff and tendon injuries	1339
	Shoulder impingement syndrome	798
	Adhesive capsulitis	165
	Shoulder lesions	139
	Posttraumatic/primary arthrosis	110
	Shoulder impingement syndrome	798
	Synovitis and tenosynovitis	102
	Arthritis	71
	Sprains and strains	58
Upper arm	Muscle and tendon rupture/injury	305
Elbow/Lower arm	Medial epicondylitis	1277
	Lateral epicondylitis	1182
	Articular cartilage disorders	101
	Lesion of the ulnar nerve	88
	Muscle and tendon injury	84
	Joint disorders	53
Wrist/Hand	Carpal tunnel syndrome	561
	Trigger finger	353
	Ligament rupture	94
	Synovitis and tenosynovitis	86
	De Quervain’s disease	63
	Sprains and strains	62
	Arthritis	57
Low back	Intervertebral disc disorder	3377
	Herniation of intervertebral disc	650
	Spinal stenosis	374
	Sprains and strains	261
	Traumatic rupture of lumbar intervertebral disc	240
	Spondylolisthesis	138
Leg/Foot	Meniscus disorder/tear, medial/lateral meniscus	1258
	Posttraumatic/primary arthrosis	340
	Cruciate ligament injury	76

**Table 2 ijerph-20-01024-t002:** Results of logistic regression analyses.

Independent Variable	*n*	OR *	95% CI	% Concordance
Industry type				42.2
Transportation/warehouse	1515	1		
and communication				
Mining	1035	2.50	2.10–2.97	
Manufacturing	15824	1.94	1.74–2.15	
Construction	4559	3.44	3.04–3.91	
Others	8240	1.16	1.04–1.29	
Size				36.5
<49	14682	1.02	0.93–1.11	
50–99	2713	1		
100–299	3561	1.11	0.99–1.23	
300–499	1397	1.06	0.93–1.22	
≥500	8820	1.32	1.20–1.45	
Gender				23.4
Women	7359	1		
Men	23814	1.59	1.51–1.68	
Age				44.2
20s	1225	1		
30s	3628	2.14	1.88–2.44	
40s	7211	2.58	2.53–3.23	
50s	11024	3.33	2.95–3.75	
60s	7357	3.10	2.74–3.51	
≥70	728	1.35	1.13–1.63	

* OR, odds ratio; CI, confidence interval.

## Data Availability

The datasets used and/or analyzed during the current study are available from the corresponding author on reasonable request.

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
