# Peer review of "Characteristics of Work-Related Musculoskeletal Disorders in Korea"

_ijerph, 2023, doi:10.3390/ijerph20021024_

Round 1
Reviewer 1 Report
This manuscript is a well-written research paper that analyzes data on musculoskeletal disorders in Korea for about 25 years, from 2016 to 2020.
Here are some comments for improving the quality of this manuscript.
1. It would be nice to add 2-3 other keywords instead of 'Musculoskeletal disorders' in keywords. (For example, low back and neck pain, trends of musculoskeletal disorders, etc.)
2. '2.1. Data collection', line 84, please add the reference in the first sentence.
3. Based on the existing data, the occurrence of musculoskeletal disorders on the shoulder was similar to or more than those on the back (see Figure 8). If you also have shoulder data, it would be nice to show the shoulder and back pain together or separately in Figure 4.
4. 'Discussion' lines 339-343, somewhat difficult to understand. It would be nice to rephrase this paragraph.
Author Response
The author appreciates your very careful, detailed and favorable review of my paper, which will be very helpful in improving it better. The author has diligently tried to fully address the issues that you indicated, which are as follows:
- It would be nice to add 2-3 other keywords instead of 'Musculoskeletal disorders' in keywords. (For example, low back and neck pain, trends of musculoskeletal disorders, etc.)
- Two keywords of low back pain and rotator cuff injury were added, and a keyword of musculoskeletal disorders was deleted.
- '2.1. Data collection', line 84, please add the reference in the first sentence.
- The reference of ‘[23]’ was added.
- Based on the existing data, the occurrence of musculoskeletal disorders on the shoulder was similar to or more than those on the back (see Figure 8). If you also have shoulder data, it would be nice to show the shoulder and back pain together or separately in Figure 4.
- As stated in the manuscript, WMSDs are classified into three representative diseases in the yearbooks of Korea: WMSDs caused by 1) bodily burdensome work, 2) chronic and accidental low back pain, and 3) carpal tunnel syndrome. So, the author has not data on shoulder WMSDs except for three year’s data obtained from COMWEL.
- 'Discussion' lines 339-343, somewhat difficult to understand. It would be nice to rephrase this paragraph.
- The paragraph has been rephrased: The incidence rate of WMSD cases per 10,000 workers in Korea was much lower than that in the USA. While the rate in the USA was 27.2 per 10,000 full-time workers in 2018 [2], the rate in KOREA was 5.1 per 10,000 workers in 2020. However, the working environment in Korea may not be safer than that in the USA. This may mean that the incidence of WMSDs might likely increase in Korea, compared to the high incidence rate of WMSDs in the USA. Therefore, preventive efforts for reducing WMSDs by the Korean government and industries are needed.
Reviewer 2 Report
The manuscript analyzed the trends of WMSD from 1996 to 2020 and investigated WMSD acceptance rates, distribution by WMSD names, effects of industry type and size, and workers' age and gender on WMSD occurrence. Compared to the previous studies focused on the prevalence of WMSDs based on subjective questionnaire surveys, this study used real WMSD data published by governmental agency in Korea that the results of the manuscript can give valuable insight to prevent WMSDs.
p.2 l55-61: The Rural Development Agency (RDA), an affiliation of the Ministry of Agriculture, ..... diseases (2.9%), and nervous system diseases (2.1%). => reference should be included.
2. Two terms ‘acceptance rates’ and ‘approval rates’ are used in the manuscript. If there is no difference, then it is suggested to use only one term.
3. Figure 1 : there is no legend for WMSDs in the graph.
4. p.12 line 266 (Logistic regression) : OR ratio of 3.5 is interpreted as “3.5 times the probability that ...”, but I think OR of 3.5 is 3.5 times the ratio of ‘# of acceptance’ to ‘# of not accepted’.
5. p.12 line 272-273 : The odds ratios did not differ significantly depending on the size, but large companies with ≥ 500 workers had the highest odds ratio. => Only large companies with ≥ 500 workers showed significantly higher OR than ‘50-99 workers’.
6. page 15, 376-385. “The chi-square tests and logistic regression analyses showed that workers' age was significantly associated with WMSDs.,..... In contrast, the present study agrees with the results of Dianat et al. [32], Heiden et al. [33], and Vieira et al. [34], which pointed out that age, gender, and length of service were associated with WMSD symptom prevalence.” => The dependent variable of logistic regression was 1 for acceptance and 0 for rejected that the results should not be directly linked to the prevalence of WMSDs. More careful expression should be used.
7. Others
1) p1 line9 : This study aimed to analyze trends for work-related musculoskeletal disorders (WMSDs) from 1996 to 2020 in Korea and (to) investigate characteristics of WMSDs
2) Fig 2 : Y axis title : incidence rate per ... => I(capital)ncidence rate per
3) Fig 5: legend MSDs => WMSDs
4) Fig 6 (a) industry => “industry type” is suggested
5) Table 2 : Age 60s 2.743.51 => 2.74-3.51
6) p14 line 310-311 : They were examined from 1996, when WMSDs were compiled as official statistics in Korea, to 2000 (=> 2020) to grasp the overall trend of WMSDs.
Author Response
The author appreciates your very careful, detailed and favorable review of my paper, which will be very helpful in improving it better. The author has diligently tried to fully address the issues that you indicated, which are as follows:
- p.2 l55-61: The Rural Development Agency (RDA), an affiliation of the Ministry of Agriculture, ..... diseases (2.9%), and nervous system diseases (2.1%). => reference should be included.
- The reference was included.
- Two terms ‘acceptance rates’ and ‘approval rates’ are used in the manuscript. If there is no difference, then it is suggested to use only one term.
- The term of ‘Acceptance rates’ has been replaced with ‘approval rates.’
- Figure 1 : there is no legend for WMSDs in the graph.
- The legend was added in Figure 1.
- p.12 line 266 (Logistic regression) : OR ratio of 3.5 is interpreted as “3.5 times the probability that ...”, but I think OR of 3.5 is 3.5 times the ratio of ‘# of acceptance’ to ‘# of not accepted’
- The author agrees with the reviewer’s comment, which has been already reflected in the original manuscript: Construction had an almost 3.5 times the probability that an applied MSD would be approved as work-related compared to the probability for the transportation/warehouse & communication industry.
- p.12 line 272-273 : The odds ratios did not differ significantly depending on the size, but large companies with ≥ 500 workers had the highest odds ratio. => Only large companies with ≥ 500 workers showed significantly higher OR than ‘50-99 workers’.
- The reviewer’s recommendation has been fully reflected in the manuscript: Only large companies with ≥ 500 workers showed significantly higher odds ratio than companies with 50‒99 workers.
- page 15, 376-385. “The chi-square tests and logistic regression analyses showed that workers' age was significantly associated with WMSDs.,..... In contrast, the present study agrees with the results of Dianat et al. [32], Heiden et al. [33], and Vieira et al. [34], which pointed out that age, gender, and length of service were associated with WMSD symptom prevalence.” => The dependent variable of logistic regression was 1 for acceptance and 0 for rejected that the results should not be directly linked to the prevalence of WMSDs. More careful expression should be used.
- The author also agrees with the reviewer’s comment. The author has changed the word of ‘WMSDs’ in the second line to ‘WMSD approval.’ A sentence was added for clarifying the author’s discussion: The chi-square tests and logistic regression analyses showed that workers' age was significantly associated with WMSD approval. The high WMSD approval rate may be interpreted as the high prevalence of WMSDs.
- Others.
- All comments that you presentd have been reflected in the manuscript.
Reviewer 3 Report
Researchers have provided a complete report of the epidemiology of the work-related musculoskeletal disorders (WMSDs) from 1996 to 2020. I have few comments:
-In the distinction between the prevalence and incidence, in the sections where the word "incidence" was used, was the “incidence” correctly estimated based on new cases?
-What is the reason that in the abstract, introduction and method, only the word “incidence” was used, but in the results section and figures, the word “incidence rate” was used? Because the way of estimating these two measures can be different. Incidence rate is a measure of incidence that incorporates time directly into the denominator (person-time).
Author Response
The author appreciates your very careful, detailed and favorable review of my paper, which will be very helpful in improving it better. The author has diligently tried to fully address the issues that you indicated, which are as follows:
- In the distinction between the prevalence and incidence,in the sections where the word "incidence" wasused, was the “incidence” correctly estimated based on new cases?
- In this study, the term of ‘incidence’ has been used for representing quantitative terms (i.e., the number of WMSD cases), while ‘prevalence’ for expressing proportion relevant terms (i.e., prevalence rate).
- What is the reason that in the abstract, introduction and method, only the word “incidence” was used, but in the results section and figures, the word “incidence rate” was used?Because the way of estimating these two measures can be different.Incidence rate is a measure of incidence that incorporates time directly into the denominator (person-time).
- Following your comment, ‘incidence’ in lines 15, 39, and 95 in the revised manuscript has been replaced with ‘incidence rate.’
Reviewer 4 Report
Thank you for opportunity for reviewing this interesting paper. The research adhere to reporting strobe guidelines. After carefully reading this manuscript, I must say that, from my point of view, the authors have done research on an important topic related with the importance of the influence of prevalence and characteristics of work-related musculoskeletal Disorders. This could be interesting clinicians, universities, private research organizations, and independent scientists, that frequently work in this area. It could give them a wider concept about and helps advance recognition of the input of different health professionals into the management of this condition, and helps inform the need for further multi-professional work in this area. This is an interesting aim with the are clinical outcomes in musculoskeletal disorders and characteristics of work . I have considered the quality of the manuscript redaction and presentation, the quality of the research methodology, the novelty and importance of the observations, and the appropriateness for the Journal’s readers (according with the Journal’s name). I have no real problems with the text of this paper, only some suggestions that are mentioned below. It appears as if the authors have done the study well, have answered an interesting clinical question with their work but there are a major concerns with the manuscript that require attention prior to publication. These will be discussed below relative to the sections of the manuscript. TITLE The title of this manuscript is very long. Perhaps a more concise version for clarity, interes and ease of read. KEYWORDS: Please use recognised MeSH terms as this will assist others when they are searching for information on your research topic. The following website will provide these (simply start typing in a keyword and see if it exists or find an alternative if it does not): https://www.ncbi.nlm.nih.gov/mesh ABSTRACt: It is hard to get the detail in an abstract when the word count is limited and this is often the hardest part of a paper to write. However, I do feel that it would be beneficial to explain the aim and conclusions what specifically you are looking at in relation to outcomes in this research. INTRODUCTION I suggest that background should be improved, with more details about prevalence of work-related musculoskeletal disorders in physical therapists, more info in the research of
prevalence of Work-Related Musculoskeletal Disorders in Iranian Physical Therapists https://pubmed.ncbi.nlm.nih.gov/30098820/ and Losa et al https://pubmed.ncbi.nlm.nih.gov/22332486/
Thus, it is indeed important paper but it lacks several critical references, in which it was presented related with this condition, and it should be emphasized in the INTRODUCTION or Discussion of the authors' paper. METHODS This section is poor, needs to present a better rationale for the study and the methodology employed. Also, neither appear information related with inclusion and exclusion criteria, dates, protocol. RESULTS The results is clear and concise with appropriate statistical analysis been performed appropriately and rigorously. DISCUSSION. This section is very short a rehashing of the results. It does not appear that the authors include much interpretation of what the study findings mean for clinical practice or research related with other research. CONCLUSION: These conclusions need to be softened, modified a in order to reflect only the study findings.
Author Response
The author appreciates your very careful, detailed and favorable review of my paper, which will be very helpful in improving it better. The author has carefully read your comments several times, and contemplate them for a couple of hours. The author has diligently tried to fully address the issues that you indicated, which are as follows:
- TITLE The title of this manuscript is very long. Perhaps a more concise version for clarity, interes and ease of read.
- The title has been changed: Characteristics of Work-related Musculoskeletal Disorders in Korea.
- KEYWORDS: Please use recognised MeSH terms as this will assist others when they are searching for information on your research topic. The following website will provide these (simply start typing in a keyword and see if it exists or find an alternative if it does not): https://www.ncbi.nlm.nih.gov/mesh
- The author tried to use recognized MeSH terms related to the manuscript. Two keywords of industrial accidents and occupational disease were found in the website that you kindly provided. However, a keyword of work-related musculoskeletal disorders was not found, but the term was still used in the revised manuscript, because it has been frequently adopted as a keyword in the field of industrial safety and health. A key of musculoskeletal disorders was deleted following other reviewer’s comment. Instead, two keywords of low back pain and rotator cuff injury, which were found in MeSH terms, were added.
- ABSTRACt: It is hard to get the detail in an abstract when the word count is limited and this is often the hardest part of a paper to write. However, I do feel that it would be beneficial to explain the aim and conclusions what specifically you are looking at in relation to outcomes in this research.
- The aim of this study has been stated in the first line of original manuscript.
- A sentence of ‘Conclusion’ was added at the end of Abstract: It can be concluded that the WMSD preventive efforts should focus on low back pain and rotator cuff syndrome by WMSD name, manufacturing by industry, small business by industry size, men by gender, and aged workers by age.
- The last sentence in the original manuscript was deleted for shortening Abstract.
- INTRODUCTION I suggest that background should be improved, with more details about prevalence of work-related musculoskeletal disorders in physical therapists, more info in the research of
prevalence of Work-Related Musculoskeletal Disorders in Iranian Physical Therapists https://pubmed.ncbi.nlm.nih.gov/30098820/ and Losa et al https://pubmed.ncbi.nlm.nih.gov/22332486/
Thus, it is indeed important paper but it lacks several critical references, in which it was presented related with this condition, and it should be emphasized in the INTRODUCTION or Discussion of the authors' paper.
- Two references that you suggested were stated in lines 53-56 of Introduction, and discussed in lines 357, 384, and 389 of Discussion.
- METHODS This section is poor, needs to present a better rationale for the study and the methodology employed. Also, neither appear information related with inclusion and exclusion criteria, dates, protocol.
- Rationale for using official yearbooks was provided in lines 88-90 of Materials and Methods. The purposes for some analyses were added in lines 94 and 99.
- All data obtained from COMWEL were used in this study, which were stated in line 122.
- RESULTS The results is clear and concise with appropriate statistical analysis been performed appropriately and rigorously.
- The author should appreciate your very favorable comment.
- DISCUSSION. This section is very short a rehashing of the results. It does not appear that the authors include much interpretation of what the study findings mean for clinical practice or research related with other research.
- The first paragraph of Discussion summarized the results of this study, which may be a rehashing of Results as you referred. However, the remaining paragraphs presented interpretations or reasons for some results of Results, and stated limitations of this study. For highlighting this point, some sentences were added in lines 361-362, 383-385, and 388-390. In addition, a sentence of lines 146-147 in Results was deleted.
- CONCLUSION: These conclusions need to be softened, modified a in order to reflect only the study findings.
- A sentence with a rather subjective meaning of 422-423 in the original Conclusions were deleted. The meanings of this study have been rewritten.
Round 2
Reviewer 1 Report
I think the author responded adequately to my all comments.
Thanks,
Reviewer 4 Report
I am happy with the paper as it stands. Congratulations.